# Antigenic Essence: Upgrade of Cellular Cancer Vaccines

**DOI:** 10.3390/cancers13040774

**Published:** 2021-02-12

**Authors:** Petr G. Lokhov, Elena E. Balashova

**Affiliations:** 1BioBohemia Inc., 177 Huntington Ave., Boston, MA 02115, USA; balashlen@mail.ru; 2Institute of Biomedical Chemistry, Pogodinskaya st., 10/8, 119121 Moscow, Russia

**Keywords:** antigenic essence, cancer vaccine, proteomic footprint, mass spectrometry, antiangiogenic vaccine, endothelial cells, preventive vaccine, SANTAVAC

## Abstract

**Simple Summary:**

Early cancer vaccines include whole-cell formulations, which operate on the principle that you should vaccinate with what you want to develop protection against. Such vaccines have been widely tested in various cancers and their advantages described but have not yet managed to pass clinical trials. Antigenic essence technology offers the possibility to revitalize the field of whole-cell-based vaccination, as the essence comprises the entire diversity of native cellular antigens. At the same time, the technology allows for precise control and purposeful change of essence composition as well as purification of essence from ballast cellular substances and also addresses issues of major histocompatibility complex restriction. Antigenic essence technology makes it possible to update many cellular vaccines that have already been developed, as well as to develop new ones, therefore introducing a new direction for anticancer vaccination research.

**Abstract:**

The development of anticancer immunotherapy is characterized by several approaches, the most recognized of which include cellular vaccines, tumor-associated antigens (TAAs), neoantigens, and chimeric antigen receptor T cells (CAR-T). This paper presents antigenic essence technology as an effective means for the production of new antigen compositions for anticancer vaccination. This technology is developed via proteomics, cell culture technology, and immunological assays. In terms of vaccine development, it does not fit into any of the above-noted approaches and can be considered a new direction. Here we review the development of this technology, its main characteristics, comparison with existing approaches, and the features that distinguish it as a novel approach to anticancer vaccination. This review will also highlight the benefits of this technology over other approaches, such as the ability to control composition, optimize immunogenicity and similarity to target cells, and evade major histocompatibility complex restriction. The first antigenic essence products, presented under the SANTAVAC brand, are also described.

## 1. Introduction

Cellular cancer vaccines are among the first types of cancer vaccines to be tested [1]. They use a straightforward approach based on the well-known principle of vaccination: vaccinate with what you want to develop protection against. Cancer cells are typically irradiated, combined with an immunostimulatory adjuvant, and then administered to the patient, usually from whom the tumor cells were isolated [2,3,4,5]. Such vaccines have undergone extensive scientific research and have been widely tested in various cancers, including lung cancer [5,6,7], colorectal cancer [3,8,9,10], melanoma [11,12,13], renal cell cancer [14,15,16], and prostate cancer [2]. The significant possibilities of such vaccines have been described, but none have yet passed the clinical trial stage.

One alternative to vaccination with cancer cells is vaccination with single tumor-associated antigens (TAAs), another mainstream approach. A wide variety of TAAs have been tested [17,18]. Among these are cancer/germline antigens (e.g., MAGE-A1, MAGE-A3, and NY-ESO-1); cell lineage differentiation antigens (e.g., tyrosinase, gp100, MART-1, prostate-specific antigen (PSA), and prostatic acid phosphatase (PAP)); and antigens that are overexpressed in cancer cells (e.g., hTERT, HER2, mesothelin, and MUC-1). Several TAAs, normal host proteins that are abnormally expressed in cancer cells, appear to be suitable targets for immunotherapies, but few of them possess the specificity or immunogenicity needed to pass clinical trials.

Another trend is neoantigens, which arise from mutated proteins in cancer cells. Neoantigens are specific to each cancer, and their diversity allows researchers to select the most immunogenic among them, potentially bypassing major histocompatibility complex (MHC) restrictions and allowing for use in immunotherapy. Major hurdles for neoantigen vaccination include the lengthy and cumbersome process necessary for the identification and selection of neoantigens, the exclusively personalized approach, and the limited number of tumors that possess enough mutations to apply this technology to (Appendix A) [19].

Overcoming the limitations of antigens could revitalize some strategies that have not yet led to final products. For example, chimeric antigen receptor T cells (CAR-T) technology uses T cells that have been genetically engineered to produce an artificial T cell receptor for use in immunotherapy [20]. The main advantage of CAR-T cells is that they bypass MHC restriction, allowing for direct activation of effector cells for the treatment of various tumors with already discovered TAAs. This opportunity to apply already-known TAAs has led to a new trend in immunotherapy.

It can be argued that antigenic essence technology is the revitalization of cellular vaccination, as antigenic essence exhibits all the identifying properties of cancer cells while also allowing for control of composition, purification from ballast substances (cellular noise), and evasion of MHC restrictions. Therefore, antigenic essence technology (i) allows for the application of already discovered cellular compositions, while (ii) overcoming their limitations, which are signs of a trend-forming technology. A comparative analysis with previous technologies, including vaccination with whole cancer cells (cellular vaccines) or single TAA, CAR-T, and neoantigens, illustrates this (Figure 1).

Regarding tumor specificity, antigenic essence technology outperforms all other methods. The high specificity of antigenic essence is provided by the similarity of the essence code to the surface antigen profile of tumor cells. In whole-cell vaccines, specificity is diluted by the presence of antigens from throughout the cell. Although it is technically more difficult to isolate surface antigens alone, it is an exceptional feature to gain and offers a major competitive advantage over whole-cell vaccines. Vaccination based on TAAs and CAR-T received a lower specificity rating due to their use of antigens that are not only overexpressed in tumor cells but also expressed in normal cells. Moreover, the specificity of response is determined mainly by the expression levels of selected antigens in tumor cells and cannot be efficiently modulated in these approaches.

In neoantigen technology, the neoantigens most suitable for vaccination are usually found in highly mutated tumors (e.g., lung cancer or melanoma) [19]. Such antigens allow for the highly specific destruction of those cells in which they were identified. However, highly mutated tumors are heterogeneous, containing clones of different cells with a different set of mutations [21]. Therefore, neoantigens may be specific to a certain cancer cell type but not to the tumor as a whole. This is considered to be a disadvantage (Figure 1).

The possibility of a tumor escaping from a vaccination-induced immune response is only partially addressed by approaches that vaccinate using individual antigens. Mutations that lead to reduced expression of the targeted antigens can allow cancer to escape the immune response. A better solution is to use whole-cell vaccines to target a broad range of antigens, thus increasing the number of mutations the tumor would have to develop in order to escape [22,23]. However, whole-cell vaccines have so far failed to address the issue of antigen modification under the selective pressure of drug treatment [24,25]. Antigenic essence technology overcomes this issue by matching antigenic essence codes with surface profiles of target cancer cells, which makes antigenic essence technology potentially the most successful with regard to tumor escape mutations (Figure 1).

Regarding MHC restriction, neoantigen and CAR-T technologies both address this issue. Only those neoantigens which are presented by MHC are selected for use in vaccination, and CAR-T technology overcomes the MHC barrier for the involved TAAs [20]. Whole-cell vaccines fail to address MHC restriction since whole cells contain a full complement of antigens, including some that evade MHC restriction (e.g., immunopeptidome [26]) and others that do not. Antigenic essence can be enriched with both immunopeptidome as well as surface antigens presented by MHC, as demonstrated by the antigenic essence product SANTAVAC.

## 2. Antigenic Essence Concept

The concept of antigenic essence arose from that of cellular vaccination, the driving principle of which is to vaccinate with what you want to destroy by vaccination. Despite the persistent failure of vaccines based on whole tumor cells, the development of such vaccines continues to occupy a leading position along with those based on TAAs [27]. Cells are a natural source for the entire diversity of native antigens, so the concept of antigenic essence takes advantage of this while also minimizing the limitations associated with the use of whole cells.

The first assumption of antigenic essence is that only antigens presented on the surface of the cell are targets for vaccination, while those presented inside the cell can and should be neglected. This is based on the fact that only antigens on the cell surface are available to the immune system; the plasma membrane of living cells is impermeable to cytotoxic elements of the immune system that recognize antigens, be they antibodies or T lymphocytes. This does not contradict the fact that some intracellular antigens can be presented by MHC on the cell surface and offer suitable targets for vaccination; indeed, some such antigens are currently undergoing clinical trials. On the other hand, extracellular antigens are produced inside cells and can therefore always be found there as well. The point is not where the antigens can be found, but that they function as targets for the immune system only when presented on the cell surface. That is, the target pool of antigens is located precisely on the cell surface, and they form an actual profile of the antigenic properties of living cells.

Given that the target pool of antigens is presented on the cell surface and the cell wall is impermeable to macromolecules, the second assumption of antigenic essence is that the target pool of antigens can be collected by treating living cells with protease (Figure 2). Despite the apparent simplicity of this approach, this is not a trivial process, and the implementation in mammalian cells became possible only with advances in proteomics. When a highly purified protease is used, cytolytic activity is low [28], thus allowing the treatment of live cells without violating the permeability of their membranes. The use of proteomics-grade protease makes it possible to obtain just the pool of antigens from the cell surface without contamination by intracellular content. Furthermore, the obtained antigens are not overridden by impurities or autolytic fragments of the protease itself [28]. Last but not least, the obtained antigens can be analyzed by mass spectrometry to establish essence composition, and this information can be used to fine-tune the design of cancer vaccines and validate their quality.

## 3. Antigenic Essence Equivalence to Cell Antigens

To confirm that antigenic essence retains the antigenic properties of live cells, its cytotoxic properties were compared with those of whole cells. In cytotoxicity assays, target human adenocarcinoma cells (MCF-7) were incubated with human cytotoxic T lymphocytes (CTLs) stimulated with antigen-presenting cells (APCs) loaded with either essence or whole cancer cell lysate. The essence was 10–40% more effective at killing target cells than the whole-cell lysate (Figure 3a). This was accomplished using less than 1% of the total amount of protein (2 μg/mL of essence vs. 270 μg/mL of whole-cell lysate) [30]. The complete retention of cytotoxicity combined with a dramatic reduction in protein used indicates that the essence is free from superfluous intracellular content and preserves all the antigenic properties required for targeting by the immune response. These results, together with a similar result reported for non-cancer cells in another study [31], confirmed the conceptual assumption that proteolytic treatment of living cells allows the collection of a complete set of antigenic targets; that is, the antigenic essence of cells.

The feasibility of anticancer vaccination with antigenic essence has also been demonstrated in a mouse model [32]. Pilot experiments in 2006 confirmed the vaccination potential of antigenic essence (Figure 3b) and made it possible to begin research and development work toward final products for vaccination.

## 4. Control of Antigenic Essence Composition

Antigenic essence is a product of cell culture technology, which involves a high probability of misidentification and cross-contamination between cells [33,34,35,36]. Moreover, cultured mammalian cells have a finite lifespan [37,38] and are subject to progressive degeneration, which is manifested in part by changes in the molecular composition of the cell surface [39,40]. Accordingly, cultivated cells intended for therapy must be authenticated and the composition of their antigenic essence validated to exclude unwanted changes in their composition.

The proven method for analysis and control of essence composition is mass spectrometry. This method is known as a proteomic footprint and represents a proteomics method for both the authentication and characterization of cells at the subtype level [29] as well as analysis of essence composition. By comparing the mass spectrum of the essence with that of the reference cells, the origin of the essence can be easily authenticated and its composition verified for vaccination (Figure 4). Moreover, discrepancies in this comparison would reveal any changes in antigen composition that may have occurred.

## 5. Antigenic Essence and MHC Restriction

### 5.1. Antigens Size in Essence and MHC

CTLs are specialized and effective elements of the anti-tumor immune response. CTLs directly lyse target cells and also secrete cytokines (e.g., granulocyte-macrophage colony-stimulating factor, tumor necrosis factor, and interferon-gamma), which further amplify immune reactivity against target cells [41,42,43]. It is well established that CTLs recognize antigens in the form of small peptides. Native whole antigens are internalized and proteolyzed by APCs, and then short peptides are presented to CTLs on the APC surface with MHC (8–12 amino acid residues in length for MHC class I, and 11–30 residues in length for MHC class II) [44].

From this point, the antigenic essence compositions are ideally suited for presentation to CTLs by MHC. While MHC class I peptides contain on average 10 residues, proteins have in their sequence an average of one trypsin cleavage site per 10 residues. This coincidence is not accidental: antigenic essence arose from proteomics technologies, in which protein cleavage with trypsin is used to produce specific peptide fragments for protein identification (known as peptide fingerprinting). Smaller fragment sizes (i.e., less than 10 residues) result in less specificity. MHC class I likely presents peptides of a similar size because they are optimal for identification. These common fundamentals of proteomics and immunology have supported the design and production of antigenic essence products, in which protein cleavage with trypsin plays a central role. Notably, in essence antigens, partial cleavage by trypsin can also produce larger peptides, which are well-suited for presentation by MHC class II.

### 5.2. Essence Composition and MHC Mediated Immunogenicity

To date, tumor antigens successfully presented to CTLs have been identified by various methods [45,46,47,48,49,50]. In most cases, CTLs derived from cancer patients were used to screen either expression gene libraries or peptides eluted from tumor cell MHC molecules. Most of these CTLs were derived from patients with a particular tumor, and as a result, the majority of defined antigens were restricted to this tumor indication. Later, an alternative approach was developed that did not require the use of patient CTLs, but instead relied on the identification of MHC-binding peptides and on novel in vitro priming protocols [51,52,53]. Specifically, peptides selected for their binding capacity to MHC molecules were tested for their ability to elicit tumor-reactive CTLs in vitro using lymphocyte cultures from normal individuals. Following this approach, several TAAs presented to CTLs were identified for melanoma tumors [54,55]. Furthermore, this approach was successfully used to identify antigens expressed in solid adenocarcinomas (including *MAGE2*, *MAGE3*, carcinoembryonic antigen (CEA), and HER-2/*neu*) [56,57,58]. These results are relevant for the development of epitope-based immunotherapy of high-incidence tumors such as breast, lung, colon, and gastric carcinomas [59]. In vitro priming of CTLs from normal individuals is a useful tool for the development of dendritic cell vaccines. Such an approach has been used in an in vitro study of DC ability to present antigens to CTLs, a process in which MHC is directly involved.

This DC vaccine development routine (combined with cell footprinting data) is an essential part of essence vaccine development (Figure 5). DCs presented essence antigens to CTLs through MHC. The target cell killing rate by activated CTLs is reflected in Equation (1) describing antigenic essence efficacy:N *=* k**r* + b,(1)

In this equation, N is the number of total viable target cells in cytotoxicity assays (representing the inverse of CTL cytotoxicity), k and b represent a contribution to the immune response independent of essence/target similarity (i.e., *r* value). This contribution is referred to as antigenic essence immunogenicity influenced by MHC restriction. In the final antigenic essence compositions, these parameters should be fine-tuned in order to strike a balance in essence/target similarity and essence immunogenicity.

### 5.3. Antigenic Essence and Immunopeptidome

In the light of the concept of antigenic essence, mention should be made of immunopeptidome: peptides associated with MHC. Immunopeptidome is produced by the proteasome degradation of intracellular proteins. Immunopeptidome analysis has become one of the essential directions of working with adaptive immunity and facilitates the development of personalized cancer vaccines. Although the antigenic essence has not been studied for enrichment with peptides from immunopeptidome, a significant fraction of it is probably included in essence. Despite the fact that the main method for obtaining immunopeptidome is elution of peptides from MHC molecules [60], it is known that soluble MHC fragments with peptide ligands circulate in the bloodstream [61], which is the result of MHC cleavage by various proteases [62,63]. Moreover, to obtain such MHC fragments, the protease (papain) was used to treat living cells [64], after which peptides from the immunopeptide released with MHC fragments were identified [65]. The similarity of this approach with the production of antigenic essence indicates a possible enrichment of the antigenic essence with peptides from the immunopeptide, which means enrichment with intracellular antigens relevant for vaccination. However, the degree of such enrichment requires further research.

Summarizing the above, it can be concluded:Antigenic essences are composed of peptides of a size ideal for presentation by MHC.Antigenic essence compositions strike an optimal balance between similarity of essence code with the surface of target cells and enrichment of MHC-restricted peptide antigens.Antigenic essences are probable to contain the intracellular antigens present in the immunopeptidome relevant for vaccination.

## 6. Antigenic Essence Products

Cancer cells are known to be able to escape the immune response, which is one of the main reasons for the current lack of cancer vaccines [66]. In antigenic essence technology, this problem requires special attention. It has been demonstrated that under the action of various anticancer drugs such as doxorubicin, tamoxifen, and etoposide, cancer cells significantly modify their surface antigen profiles [25]. Repeated treatment of cells with etoposide with a dose of IC96 changed the cell phenotype so markedly that the antigenic profile deviated from the initial profile as much as the antigenic profile differs across different cell types [25]. The adaptability of essence technology allows the essence composition to be purposefully modified to match the altered surface profile of the target cancer cells. The technology developer has further singled out endothelial cells (ECs) as strategic targets for antiangiogenic essence since ECs are more genetically stable and therefore less prone to develop escape mutations than cancer cells [67].

In the context of cancer, ECs have been studied extensively [68]. In 1945, it was reported that in an animal model, a grafted tumor recruits the host’s capillaries to support its nutrition and growth [69]. This discovery gave rise to a whole field of research aimed at suppressing the formation of new blood vessels [68]. Wei et al. [70] conducted the first vaccination study using whole EC preparations to target the tumor vasculature. It was confirmed that the vaccination of mice with xenologous ECs inhibited tumor growth. In addition, Okaji et al. [71] observed anti-EC immune responses in mice that were immunized with fixed autologous ECs. Immunogenicity assays showed that both cross-reactive antibodies and CTLs were induced upon immunization. In 2006, Chen et al. [72] developed a vaccine based on viable ECs and then used it to vaccinate against myeloma cells and Lewis lung carcinoma cells, demonstrating that tumor growth can be inhibited by antibodies or CTLs induced by vaccination. In 2008, the first clinical results of this vaccine were published [73]. Glutaraldehyde-fixed human ECs induced a specific anti-EC antibody response in 8 out of 9 patients, while only six patients induced a specific cellular response. Three patients with malignant brain tumors showed a partial or complete response. These data form the basis for further developments and updates of EC-based vaccines.

The essence-based upgrade of EC-based cancer vaccines, under the brand SANTAVAC (a Set of All Natural Target Antigens for Vaccination against Cancer), were produced from human microcirculatory endothelial cells (HMECs) derived from fat tissue. The mechanism of action of SANTAVAC is based on EC heterogeneity and consists of the use of peptide profiles (essence code), which is similar to the surface antigens of tumor vessels, that allows targeted destruction of tumor vessels by vaccination. The use of proteomic footprinting and cytotoxicity assays to develop such essence compositions is described in detail in several publications (Figure 6a) [74,75,76,77]. The HMEC model made it possible to purposefully change the profile of surface antigens and obtain antigenic essence from them, demonstrating in vitro the high efficiency of the essence in comparison with other cellular preparations [75].

While additional mechanistic and functional studies using SANTAVAC are ongoing, the in vitro results demonstrate how antigenic essence technology contributed to the development of allogeneic SANTAVAC as a vaccination candidate with increased efficacy and safety. The SANTAVAC formulation achieved efficacy equal to 17 and 60 in relation to in vitro predictions of vaccine safety and capacity to arrest tumor growth, respectively [75]. These data directly defined compositions of final SANTAVAC products. SANTAVAC^15^ is a candidate for therapeutic cancer vaccines (Figure 6b). Although SANTAVAC^25^ exhibited less strength of action than SANTAVAC^15^, it is also being considered as a candidate for cancer vaccines. It is expected that SANTAVAC^25^ will not affect vasculature in normal tissues or vasculature weakly stimulated by the tumor, resulting in very low side effects. This suggests that it may be a suitable candidate for preventive cancer vaccines, which are used in disease-free people and thus have stricter safety requirements.

The SANTAVAC data provide the first positive example of the effectiveness of technology in designing antigenic compositions for cancer vaccines. Notably, various essence candidates showed up to a ten-fold difference in cytotoxicity, including some that did nothing at all [75]. The ability to systematize these data, identify consistent patterns, and validate experimentally the most effective compositions gives antigenic essence technology a significant advantage over the use of whole-cell lysate. Considering that many whole-cell vaccines nonetheless succeed in the initial stages of clinical trials, the possibility of passing them through all stages of trials after optimization with essence technology seems scientifically sound and realistic.

## 7. Upgrade of Developed Cellular Cancer Vaccines

Some additional recommendations, based on the antigenic essence concept, can be given for upgrading developed cellular vaccines. It is important to consider the cause of their failure in clinical trials. Often, the primary endpoint of improving overall survival (OS) or similar was not met, and in this context, vaccine dosage is likely the issue. For example, the median OS of the GVAX cancer vaccine was 23 months (low-dose) and 35 months (high-dose; equal to 3× low-dose) [78]. This indicates the dose-dependent nature of cell-based vaccination, as 3× dose led to a sufficient increase in OS. While not exceeding the total protein amounts of existing vaccines and also avoiding the side effects associated with cellular inoculation, antigenic essence allows up to 100× dose in terms of the number of cells used for vaccination. Thus, it can be expected that antigenic essence would greatly increase the efficiency of existing vaccines.

For anticancer vaccinations, genetically modified cell lines are often used. Modification of these cells may increase their expression of oncoantigens, and given the findings of antigenic essence technology, they would likely be most efficient when those antigens are on the cell surface. Thus, for example, the BriaVax (BriaCell Therapeutics Corp., West Vancouver, BC, Canada) vaccine is prepared from a human breast cancer cell line that overexpresses the cell surface protein *Her2*/*neu*, which is also overexpressed in epithelial cancers [79]. This and similar cell lines are particularly amenable to upgrade since the antigens to be enriched include cell surface proteins.

Since they are acellular formulations, upgraded vaccines have undeniable advantages concerning safety. They are free of any cellular elements and supramolecular formations. The production process excludes the presence of bacteria, viruses, protozoa, as well as prions and other foreign proteins (e.g., bovine proteins from the culture medium). Thus, they are less likely to cause infection or allergic reaction. This aligns with current guidance regarding the development of new cancer vaccines with enhanced safety and tolerability profiles [80] and simplifies the implementation of Food and Drug Administration (FDA) requirements [81]. The absence of side effects will significantly reduce the door-to-needle time, which is extremely important for mass vaccination and has been a significant limiting and price-forming factor of cellular vaccines.

A significant simplification in the transportation and storage of upgraded vaccines is expected in comparison with their cellular analogs. Storage and transportation of cell preparations are usually carried out in liquid nitrogen with cryoprotectant, which is labor-intensive, and the defrosting of cells is typically associated with massive cell lysis. Antigenic essence, being a peptide preparation, requires ultra-low temperatures only during long-term storage [82], and in general, can be frozen normally or, as a lyophilizate, stored at low temperatures (+4–8 °C) and transported at room temperature.

Radical simplification in transportation and storage, increased safety, and decreased door-to-needle time will undoubtedly lead to a radical reduction in the cost of upgraded vaccines. The cost of upgraded vaccines will also be affected by the near-absence of cellular material consumption during their production. A single set of cells can be reused to generate multiple batches of antigenic essence due to the regeneration of cells after treatment with protease [83]. Thus, virtually any dose can be generated without wasting cellular material, which is important for the mass production of affordable cancer vaccines. A notable exception to this is perhaps in upgraded vaccines obtained from primary cultures where long-term cultivation of cells without degeneration of their molecular phenotype is impossible.

## 8. Conclusions

Despite the persistent failure of vaccines based on tumor cells, the development of such vaccines continues to occupy a leading position along with TAA-based vaccines. They use a straightforward approach based on the well-known principle of vaccination: vaccinate with what you want to develop protection against. Technology for producing antigenic essence allows other promising cell-based vaccines to be revitalized and developed into new vaccine compositions.

The direct connection that has been demonstrated between essence composition and immune response, as well as its enormous potential for increasing specificity and immunogenicity, make antigenic essence technology worthy of attention. The developers of this technology count on the attention of the scientific community to provide a qualified assessment, point out shortcomings, and, in case of a positive assessment, contribute to its improvement and wide implementation into practice.

## Figures and Tables

**Figure 1 cancers-13-00774-f001:**
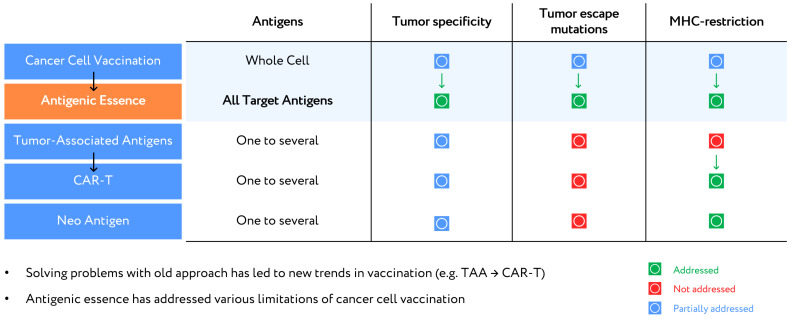
Anticancer vaccination approaches.

**Figure 2 cancers-13-00774-f002:**
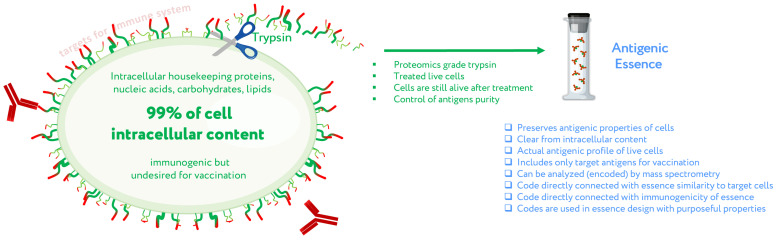
Concept of antigenic essence. Actual antigenic properties of live cells are defined by a pool of antigens presented on the cell surface. Intracellular content is considered noise to be excluded from the essence. After washing away traces of culture medium, cells are treated with a purified protease. Released fragments of the cell surface proteins are collected, analyzed by mass spectrometry, and used for vaccination instead of whole cells. Adapted from [29].

**Figure 3 cancers-13-00774-f003:**
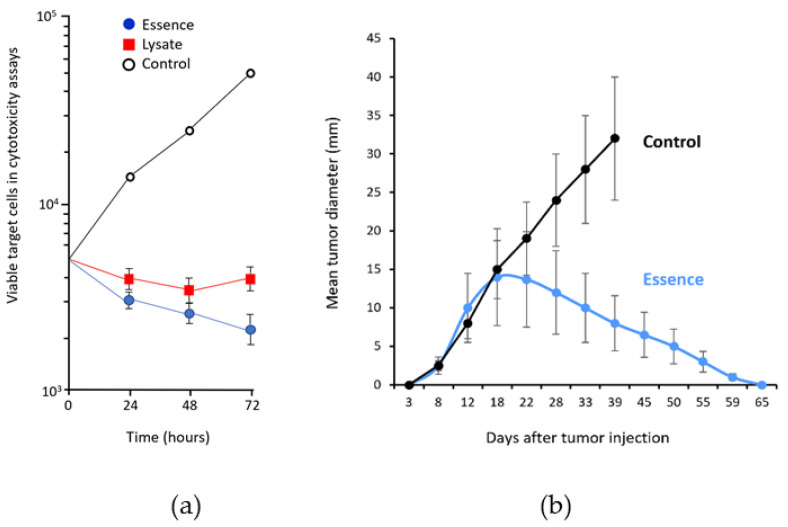
Antigenic equivalence of antigenic essence to whole cell lysate. (**a**) Cytotoxicity of effector cytotoxic T lymphocytes (CTLs) against human adenocarcinoma (MCF-7) cells. Target MCF- 7 cells were seeded in tissue culture plates, and then effector CTLs were added at CTLs:MCF-7 ratio 4:1. (●) MCF-7 cells incubated with CTLs that had been stimulated with antigenic essence-loaded dendritic cells (DCs). (■) MCF-7 cells incubated with CTLs that had been stimulated with lysate-loaded DCs. (○) MCF-7 cells grown with unstimulated CTLs. Points represent the mean value of three identical measurements. Essence induces 10–40% more cytotoxic activity in CTLs than whole cell lysate, even though the total protein concentration of essence formulation was substantially lower (whole cell lysate: 270 µg/mL vs. essence: 2 µg/mL). Adapted from [30]. (**b**) Growth of hepatoma H22 on BALB/c mice in the experimental group vaccinated with the essence of H22 cells as compared to unvaccinated controls (mean ± SD). Adapted from [32].

**Figure 4 cancers-13-00774-f004:**
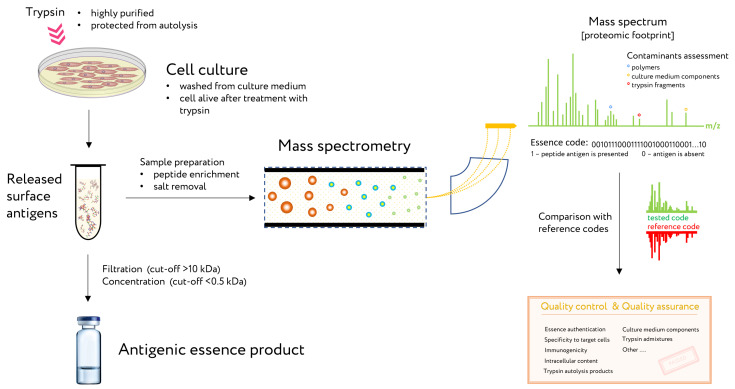
The use of proteomic footprinting in the antigenic essence production. After washing away traces of culture medium, adherent cell culture is treated with a protease. Released fragments of cell surface proteins are collected and analyzed by mass spectrometry. The set of obtained peptide molecular weights represents the essence code. Comparison of this code with mass spectra of the reference cells not only allows for authentication of the essence but also reveals any changes in its composition. This proteomic footprinting method was developed as a part of research and development for antigenic essence products and cell authentication at the subtype level. Adapted from [29].

**Figure 5 cancers-13-00774-f005:**
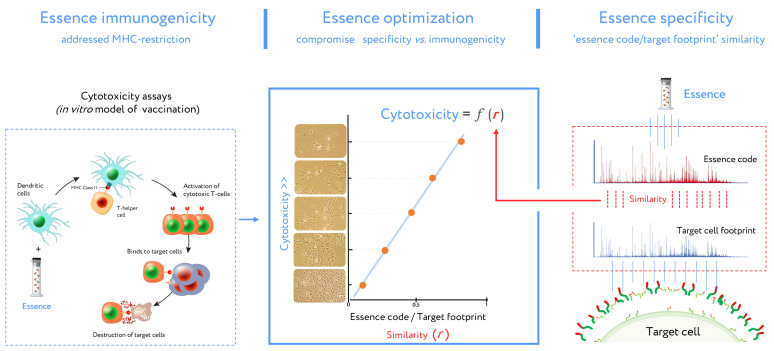
Stages of design for antigenic essence compositions. This vaccine development routine combines cytotoxicity assays (left plot) and target cell footprinting (right plot). In cytotoxicity assays, the dendritic cells (DCs) present essence antigens to CTLs through major histocompatibility complex (MHC). The target cell killing rate (cytotoxicity) by activated CTLs is reflected in the function describing antigenic essence efficacy from the essence/target similarity (i.e., *r* value). Selection of the final antigenic essence compositions involved an optimal combination of essence/target similarity and essence immunogenicity influenced by MHC restriction. Antigenic essence compositions should strike an optimal balance between similarity of essence code with the surface of target cells and enrichment of MHC-restricted peptide antigens.

**Figure 6 cancers-13-00774-f006:**
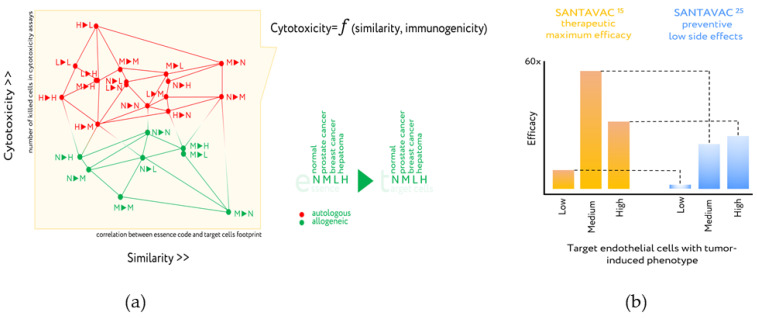
Design of antiangiogenic essence candidates based on comparison of essence code with target cell footprint and cytotoxicity assays. (**a**) Cytotoxicity is plotted vs. essence/target similarity as determined by correlation of essence composition with cell surface profile of target cells. Legend: 1►2—First letter (1) corresponds to essence, second letter (2) corresponds to the target human microcirculatory endothelial cells (HMECs) used in the cytotoxicity assay. Letters are used to identify the presence in the growth medium of either normal EC growth supplement (N), MCF-7 cell-conditioned medium (M), LNCap cell-conditioned medium (L), or HepG2 cell-conditioned medium (H). All data in the plot were described by a set of linear functions from the essence/target similarity, in which the coefficients were also linearly interdependent and attributed to the essence immunogenicity. Adapted from [74,77]. (**b**) In vitro efficacy of optimized SANTAVAC compositions for final products. Target HMECs were incubated in the presence of effector CTLs at a 1:20 ratio. After 3 days, CTLs were removed and target cell viability was determined. Efficacy was calculated as a ratio of the number of tumor-stimulated cells in control (i.e., Low: HMEC^5%^, Medium: HMEC^15%^, or High: HMEC^25%^) to the number of tumor-stimulated cells in the experiment. Thus, efficacy predicts the therapeutic effect of the vaccine in vitro. For efficacy calculations, the data representing the mean value of three independent measurements were used. Percentage values indicated in the superscript correspond to the percentage of tumor-conditioned medium used to stimulate target HMEC, or HMEC used to generate SANTAVAC. SANTAVAC^15^ corresponds to antigenic essence obtained from HMEC^15%^. SANTAVAC^25^ corresponds antigenic essence obtained from HMEC^25%^. Adapted from [75].

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
