# Peer review of "Antigenic Essence: Upgrade of Cellular Cancer Vaccines"

_cancers, 2021, doi:10.3390/cancers13040774_

Round 1
Reviewer 1 Report
I thank the authors for answering to my comments and questions despite my initial decision to reject this article. I understand your comments and reviewed the revised version of your article, however I cannot give a positive decision for the following two main reasons:
- the concept of ‘antigenic essence’ at the origin of the SANTAVAC product from BIOBOHEMIA was already established in 2016, e.g. the following citation from Lokhov PG and Balashova EE (Allogeneic Antigen Composition for Preparing Universal Cancer Vaccines. J Immunol Res. 2016;2016:5031529):
it was shown that the "antigenic essence" of cells, which may be used in cell-based vaccines in contrast to whole cells, could be prepared by proteolytic cleavage of cell surface targets. The composition of this "antigenic essence", which was established by the proteomic footprinting, directly defined target cell killing rates in CTA that represent an in vitro anticancer vaccination model. The "antigenic essence" prepared for vaccines designed to target the tumor vasculature gave rise to the name SANTAVAC.
And although the authors have taken note of erasing the redundancies observed in the figures of the current article with the figures of the previous articles, in particular in Lokhov et al. Vaccine 2019, this is not sufficient to shed light on new insight into, new ways of looking at existing knowledge and concepts that could lead to new research questions. - Trying to position the antigenic essence method with regard to other types of anti-cancer vaccines and treatment with CAR-T cells is an excellent idea, however the current format, in particular the Figure 1, does not support enough the strengths and weaknesses of each method as references to literature are lacking. The reader would expect deeper discussions as to the differences between the methods from a tumor specificity, tumor escape, MCH restriction or off-target toxicity point of view, but also other technical and logistic aspects such as the time-to-needle, ease and costs of manufacturing and product stability. Finally the reader gets confused in part 6 of this article which presents SANTAVAC which seems to be a vaccine directed not against the cancer cell but against the tumoral vasculature. Thus any comparison with methods using whole cancerous cell extracts, tumor-associated antigens or neoantigens is no longer possible.
Reviewer 2 Report
The authors have addressed my previous comments and although there is is still a high amount of self-citation, I appreciate that it is a small field and effort has been made to adapt previous figures that had been copied from previous papers.
Reviewer 3 Report
The authors addressed all raised points from the previous draft and I can therefore recommend publication.
This manuscript is a resubmission of an earlier submission. The following is a list of the peer review reports and author responses from that submission.
Round 1
Reviewer 1 Report
Summary of the main findings of the study
The review article by authors Lokhov and Balashova is an update of the knowledge acquired around the concept of Antigenic Essence which is described as an improvement in whole cell vaccination formuation, in particular in terms of control of the composition in antigenic substance but also molecular fingerprint of a state of antigenic composition a given time or condition. This article is well prepared and the concept of Antigenic Essence is in no way questioned here, however the lack of concrete novelty compared to a 2019 article in Vaccines by the same authors prevents it from claiming acceptance for publication in Cancers.
Comments on the article
Importants comments and observations
-
There is a lot of material that can be found very similar if not identical to the figures already published in “Lokhov, P.G.; Mkrtichyan, M.; Mamikonyan, G.; Balashova, E.E. SANTAVACTM: Summary of Research and Development. Vaccines 2019, 7, 186.”, such as:
current article Lokhov et al. Vaccine 2019 Figure 3A Figure 1A Figure 4 Figure 3 Figure 7B Figure 2 Figure 5 (center) Figure 4 (top) Figure 6 (left) Figure 5A
Minor points
- One of the important points which would have deserved to be discussed in more detail, especially with the dramatic acquisition of knowledge in the field on immuno-oncology and neoantigen cancer vaccine in recent years, is the possibility that several self antigens, which would a priori be included in the Antigenic Essence even if in a ‘purified’ format compared to a whole cell lysate, have deleterious effects. Indeed, these antigens are not prioritized by in silico and / or in vitro immunological tools such as the neoepitopes, could induce not an anti-tumor immunity but rather a tolerogenic response. Thus the interpretation of the comparative table in Figure 1 could be different, with in particular a tumor specificity which is significantly greater with the neoantigens than with the Antigenic Essence which contains more self peptides. In addition Tumor Escape Mutations are completely addressed to the reverse of what indicated.
Reviewer 2 Report
This review on antigenic essence was a captivating read of a relatively novel technology within cancer vaccines.
Overall the manuscript was very well written and well structured. The quality of the research contained comprehensive referencing to relevant and timely publications.
One question I had - was if all surface molecules are being cleaved from the target, processed and then presented by MHC to CTL, how are off-targets effects avoided since some proteins are not unique to a single cell type. Perhaps this could be clarified for the reader?
The figures were very comprehensive. I did find that figure 3a and 6 first column have been self plagerised. Figure 3a is a previously published original research result from this manuscript (doi: 10.4172/2157-2518.1000103) that had already been published in this review (as an adapted figure but it is not adapted is identical): doi: 10.3390/vaccines7040186.
Figure 6 first column is not adapted but an identical image (with the slight change of labeling within the graph) from this review: doi:10.3390/vaccines7040186 which again is listed as an adapted figure from these original articles but is identical. doi: 10.4161/hv.22828, doi: 10.1080/21645515.2015.1011022
Figure 4 and figure 7b again are 'adapted' from this review: doi: 10.3390/vaccines7040186 but again is much to similar and can be regarded as self-plagerism.
The large amount of self-plagerism is then leading to inappropriate self-citations.
New figures must be generated containing novel images, not reproduced or slightly amended. Having the majority of the figure unchanged constitutes plagerism.
Reviewer 3 Report
This concept paper presents an interesting novel approach to revive the field of whole cell-based anti-cancer vaccines. This approach is essentially based on the isolation of only cell-surface components of malignant or malignant-associated cells (called “essence”) to use as antigen source for vaccination purposes. According to the authors (see below for further comments), the main advantage of this technology is the ability to enrich the antigenic source for only those antigens really sensed by the immune system (i.e. surface exposed) and neglect intracellular material that the immune system is not usually able to target. The manuscript is sufficiently innovative and clear and we recommend publication. However, the following points should be first addressed or clarified:
- page 2 line 53-4: Major hurdles for neoantigen vaccination should be briefly stated here (e.g. lengthy and cumbersome process for selection/identification; exclusively personalized approach)
- page 3 line 82-84: without further explanations/indications I believe the contradiction still stands and is not resolved. Intracellular antigens still constitute a valid antigen source through antigen processing and MHC-I presentation. I infer that these antigens are still present and not excluded in the “essence” approach through the inclusion of MHC-I bound peptides present at the cell surface during the “essence” preparation. If this is the case this should be clearly stated. Otherwise it may seem that only whole protein surface antigens are considered in the essence approach without any consideration for intracellular antigens.
- page 3 line 96-7: How is this achieved? How did the authors make sure and verified that the protease is excluded from essence preparation? I think this is an essential aspect worthy of further clarifications.
- page 6 line 209-11: I believe that the above text did not demonstrate that “antigen essence compositions strike an optimal balance …”. In the above text the authors describe a linear model in which different parameters are defined that contribute to cytotoxicity. As also stated in the Figure 5 legend (page 7 lines 221-22) these parameters should be carefully assessed in order to find an “optimal balance”. The paragraph should be toned down and modified accordingly (e.g. “These parameters should be fine tuned in order to strike a balance …”).
- paragraph 6: In the approach with endothelial cells did the authors observe any cross reactivity and induced aberrant immune responses against healthy endothelial cells? I think this is an important aspect and a potential drawback of this approach.
Minor edits:
Figure 1: Neo-antigen/Tumor specificity has both blue and green box indication, I believe only one should be indicated (either “addressed” or “partially addressed”).
Page 7 line 235: reference 89 is not present in the bibliography.
The Supplementary table is not easy to read. I would advise to introduce lines to separate the different fields in the table and to avoid centered or justified text style.